# Hyperspectral Image Classification Using Custom Spectral Convolutional Neural Networks (CSCNNs)

Stephanie Rouamba
*Department of Electrical and Computer Engineering*
*University of the District of Columbia*
Washington, D.C., USA
sougrenonma.rouamba@udc.edu

Nian Zhang[*]
*Department of Electrical and Computer Engineering*
*University of the District of Columbia*
Washington, D.C., USA
nzhang@udc.edu

Wagdy Mahmoud
*Department of Electrical and Computer Engineering*
*University of the District of Columbia*
Washington, D.C., USA
wmahmoud@udc.edu

Lara Thompson
*Department of Mechanical Engineering*
*Biomedical Engineering Program*
*University of the District of Columbia*
Washington, D.C., USA
lara.thompson@udc.edu

Max Denis
*Department of Mechanical Engineering*
*University of the District of Columbia*
Washington, D.C., USA
max.denis@udc.edu

Tolessa Deksissa
*Water Resources Research Institute*
*University of the District of Columbia*
Washington, D.C., USA
tdeksissa@udc.edu

*Abstract*—**Custom spectral convolutional neural networks (CSCNNs) combine the strengths of convolutional neural networks with specialized spectral processing, resulting in improved classification accuracy by effectively capturing subtle variations in hyperspectral data. This paper proposes a CSCNNs based approach to classify hyperspectral images. The proposed method leverages the high-dimensional spectral data inherent in hyperspectral images, employing convolutional layers specifically designed to capture spectral-spatial features. By reducing dimensionality through principal component analysis and creating image patches, the model is trained to recognize complex patterns across different spectral bands. In addition, a comprehensive analysis of CSCNN performance is carried out, focusing on its architecture, key features, and benefits in computational efficiency and spectral representation. Experimental results on datasets such as Salinas-A, Pavia University (Pavia-U), and Indian Pines demonstrate that the CSCNN model surpasses traditional methods, achieving higher classification accuracy and more robust performance metrics like overall accuracy (OA), average accuracy (AA), and Kappa coefficient.**

*Keywords— deep learning, custom spectral convolutional neural networks (CSCNNs), pixel-based image classification*

## I. INTRODUCTION

Hyperspectral imaging captures a wide spectrum of light for each pixel in an image, allowing for the identification of materials, objects, and conditions that are not visible to the naked eye or detectable with traditional imaging techniques. In hyperspectral images, each pixel is considered a high-dimensional vector, with each component representing spectral reflectance at a specific wavelength. This detailed spectral information allows for the differentiation of subtle spectral variations, making Hyperspectral imaging valuable across numerous applications [1]– [12]. According to recent studies [13], hyperspectral image classification assigning each pixel to a specific class based on its spectral properties is a highly active research area within the hyperspectral community and has gained significant attention in the remote sensing field.

Unlike standard RGB images, which capture light in three bands (red, green, and blue), hyperspectral images can capture hundreds of contiguous spectral bands, offering rich and detailed information about the scene. This high-dimensional data has proven valuable in various fields, including agriculture, environmental monitoring, mineral exploration, and military surveillance. However, the vast amount of data in hyperspectral images presents challenges in processing and analysis. In fact, there are two primary challenges in hyperspectral image classification: the high spatial variability of spectral signatures and the imbalance between the limited number of training samples and the high dimensionality of the data [14]. The first challenge arises from several factors, including changes in lighting, environmental conditions, atmospheric effects, and temporal variations. The second challenge creates ill-posed problems for certain methods and reduces the classifiers' ability to generalize effectively.

The conventional methods for analyzing hyperspectral images are pixel-based approaches, where each image pixel is classified based on its spectral information [15]. This traditional approach to classification relies on the pixel because the pixel serves as the fundamental unit of satellite imagery.

Custom spectral convolutional neural networks (CSCNNs) offer several advantages over traditional pixel-based

- Corresponding author

approaches, particularly in the context of hyperspectral data analysis: 1) Spectral-Spatial Feature Integration: CSCNNs can simultaneously capture both spectral and spatial features, providing a more comprehensive understanding of the data. Traditional pixel-based approaches often treat pixels independently, missing the spatial context that CSCNNs can incorporate. 2) Improved Classification Accuracy: By leveraging deep learning techniques, CSCNNs can model complex relationships within the data, leading to higher classification accuracy compared to traditional methods that rely solely on spectral information. 3) Noise Reduction: CSCNNs can better handle noise in hyperspectral images. Traditional pixel-based methods may classify noisy or outlier pixels incorrectly, while CSCNNs can mitigate this by considering spatial information, leading to more robust classifications. 4) Dimensionality Reduction: CSCNNs can effectively reduce the dimensionality of hyperspectral data by learning lower-dimensional representations, making the processing more efficient without losing critical information. Traditional pixel-based methods may struggle with the high dimensionality of hyperspectral data. 5) Scalability and Efficiency: CSCNNs are designed to handle large-scale data efficiently, often requiring fewer computational resources as they can exploit the spectral structure more effectively than traditional methods, which may require more intensive processing for similar tasks. 6) Enhanced Spectral Representation: Custom spectral convolutions in CSCNNs are tailored to better capture the unique spectral signatures of different materials, leading to more precise and context-aware classifications than pixel-based methods, which may not fully utilize the spectral richness of the data. Therefore, it is imperative to develop a novel Custom Spectral Convolutional Neural Networks (CSCNNs) method for hyperspectral image classification.

The remainder of the paper is organized as follows: Section II details the proposed methodology. Section III delves into time series prediction using the LSTM network. Section IV provides the experimental results. Section V concludes the paper.

## II. PROPOSED METHODS

Hyperspectral image classification is a complex task due to the high dimensionality of the data and the variability of spectral signatures across different materials. To address these challenges, we propose a Spectral Convolutional Neural Network (CSCNN), to effectively learn and classify hyperspectral images.

### A. CSCNN Architecture

CSCNN combines the advantages of Convolutional Neural Networks (CNNs) with the characteristics of spectral domain processing. Its basic architecture typically includes multiple convolutional layers, activation layers, pooling layers, and fully connected layers. One of the key features of CSCNN includes spectral feature extraction. CSCNN performs convolution operations in the spectral domain, effectively capturing subtle variations and features in hyperspectral

images. Compared to traditional spatial domain processing, this approach better preserves spectral information. Another key feature of CSCNN is the layer count and parameter tuning. CSCNN 's layer count can be flexibly adjusted based on requirements, such as using a custom 17-layer network. The parameter settings of each layer directly influence the network's learning ability and classification performance.

The proposed network architecture for hyperspectral image classification leverages a custom Spectral Convolutional Neural Network (CSCNN) specifically designed to handle the high-dimensional data inherent in hyperspectral images (HSI), as shown in Fig. 1. The CSCNN architecture consists of several layers that are tailored to extract spectral and spatial features effectively, enabling the classification of each pixel in an HSI into a specific class based on its spectral signature.

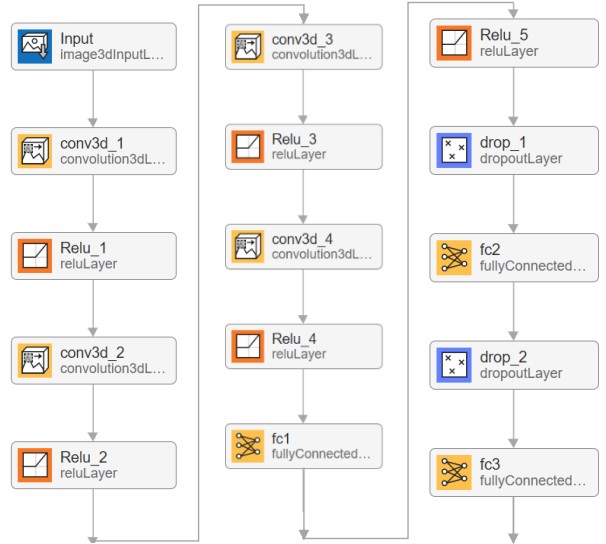

Fig. 1. The CNN layers.

### B. Input Layer

The network begins with a 3D input layer that accepts hyperspectral image patches. The input size is defined as [25, 25, 30], where 25 x 25 represents the spatial dimensions of the image patch and 30 represents the number of spectral bands after dimensionality reduction using techniques like PCA.

### C. 3D Convolutional Layers

The network includes a series of 3D convolutional layers that apply 3D filters to the input data. These layers are designed to learn spatial-spectral features by convolving across both spatial dimensions (height and width) and the spectral dimension (depth). The network typically includes multiple convolutional layers with varying filter sizes, such as:

- First 3D convolutional layer with a filter size of [3, 3, 7] and 8 filters, followed by a ReLU activation function.
- Second 3D convolutional layer with a filter size of [3, 3, 5] and 16 filters, followed by a ReLU activation function.

- Subsequent 3D convolutional layers further refine the feature extraction with progressively smaller filter sizes, capturing detailed spatial-spectral relationships.

### D. Fully Connected Layers

- Following the convolutional layers, the network includes fully connected layers to aggregate the features extracted from the convolutional layers. These layers transform the 3D feature maps into a 1D feature vector for classification purposes. Typically, there are multiple fully connected layers with a decreasing number of neurons:
- The first fully connected layer with 256 neurons, followed by a ReLU activation.
- A second fully connected layer with 128 neurons, incorporating dropout to prevent overfitting.
- A final fully connected layer corresponding to the number of classes in the dataset, followed by a softmax activation function to produce class probabilities.

### E. Output Layer

The output layer of the network uses a softmax function to generate a probability distribution across all the classes, allowing for the classification of each input patch into one of the predefined classes.

### F. Dropout Layers

Dropout layers are interspersed between fully connected layers to mitigate overfitting by randomly setting a fraction of input units to zero during each forward pass, which helps improve the model's generalization capability.

## III. EXPERIMENTAL RESULTS

In this section, we present the experimental results obtained from using the Convolutional Spectral Convolutional Neural Network (CSCNN), for hyperspectral image classification. We evaluate the effectiveness of our proposed method on the widely used hyperspectral datasets, including Indian Pines, Salinas, and Pavia University datasets.

### A. Datasets

In the experiments, we assess the proposed CSCNN network on three widely used hyperspectral scenes named Salinas-A [16], PaviaU [17], and Indian Pines [18].

- **Salinas-A**: A sub-scene from Salinas Valley, USA, captured by the AVIRIS sensor with a spatial resolution of approximately 3.7 m/pixel. The scene covers 86 lines by 83 samples and includes six classes and background elements, such as bare soil, vegetables, and vineyard grounds. The original and ground truth images are shown in Fig. 2. It has 6 ground truth classes, as shown in TABLE I.

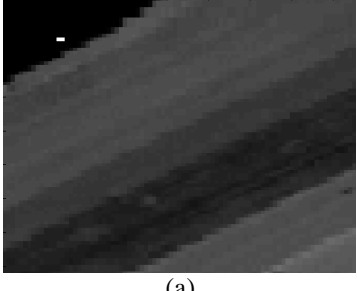

(a)

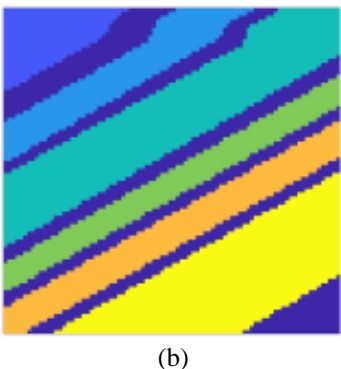

(b)

Fig. 2. (a) Salinas-A and (b) the ground truth images.

TABLE I. THE SALINAS-A CLASSES AND THEIR RESPECTIVE GROUND TRUTH SAMPLE NUMBER

| # | Class | Samples |
|---|---|---|
| 1 | Brocoli_green_weeds_1 | 391 |
| 2 | Corn_senesced_green_weeds | 1343 |
| 3 | Lettuce_romaine_4wk | 616 |
| 4 | Lettuce_romaine_5wk | 1525 |
| 5 | Lettuce_romaine_6wk | 674 |
| 6 | Lettuce_romaine_7wk | 799 |

- **Pavia-U**: Captured in 2003 over Northern Italy by the ROSIS sensor, with dimensions of $610 \times 340 \times 103$ pixels. It contains nine ground truth classes. The original and ground truth images are shown in Fig. 3. It has 9 ground truth classes, as shown in TABLE II.

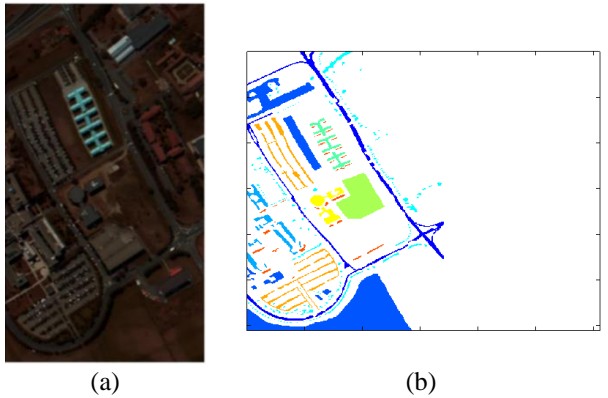

(a)        (b)

Fig. 3. (a) Pavia-U and (b) ground truth images.

TABLE II. THE PAVIAU CLASSES AND THEIR RESPECTIVE GROUND TRUTH SAMPLE NUMBER

| # | Class | Samples |
|---|---|---|
| 1 | Asphalt | 6631 |
| 2 | Meadows | 18649 |
| 3 | Gravel | 2099 |
| 4 | Trees | 3064 |
| 5 | Painted metal sheets | 1345 |
| 6 | Bare Soil | 5029 |
| 7 | Bitumen | 1330 |
| 8 | Self-Blocking Bricks | 3682 |
| 9 | Shadows | 947 |

- **Indian Pines**: Acquired over Northwestern Indiana by the AVIRIS sensor with a wavelength range of 0.4–2.5 μm, this scene comprises 145 x 145 pixels and includes 16 classes representing agricultural, forest, and road areas. The original and ground truth images are shown in Fig. 4. It has 16 ground truth classes, as shown in TABLE III.

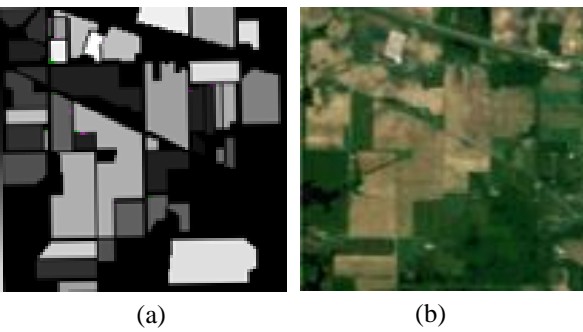

(a)                          (b)

Fig. 4. (a) Indian Pines and (b) ground truth images.

TABLE III. THE INDIAN PINES CLASSES AND THEIR RESPECTIVE GROUND TRUTH SAMPLE NUMBER

| # | Class | Samples |
|---|---|---|
| 1 | Alfalfa | 46 |
| 2 | Corn-notill | 1428 |
| 3 | Corn-mintill | 830 |
| 4 | Corn | 237 |
| 5 | Grass-pasture | 483 |
| 6 | Grass-trees | 730 |
| 7 | Grass-pasture-mowed | 28 |
| 8 | Hay-windrowed | 478 |
| 9 | Oats | 20 |
| 10 | Soybean-notill | 972 |
| 11 | Soybean-mintill | 2455 |
| 12 | Soybean-clean | 593 |
| 13 | Wheat | 205 |
| 14 | Woods | 1265 |
| 15 | Buildings-Grass-Trees-Drives | 386 |
| 16 | Stone-Steel-Towers | 93 |

## B. Experimental Setup

Each hyperspectral image was normalized and divided into smaller patches of size 25x25 pixels to ensure uniform input to the deep learning model. Dimensionality reduction was performed using PCA to reduce the number of spectral bands to 30, maintaining the most significant features. In addition, the network was trained using the Adam optimizer with an initial learning rate of 0.001, batch size of 256, and a maximum of 100 epochs. Moreover, the datasets were randomly split into training (70%) and test (30%) subsets to evaluate the model's performance on unseen data.

## C. Performance Evaluation Metrics

The performance is evaluated by the Overall Accuracy (OA), Average Accuracy (AA), and Kappa Coefficient [19]-[22]. The OA is calculated as the average of Producer's Accuracy and User's Accuracy.

$$\text{Producer's Accuracy} = \frac{\text{correctly identified pixels}}{\text{total number of pixels/class}}$$

$$\text{User's Accuracy} = \frac{\text{correctly identified pixels}}{\text{correctly identified pixels} + \text{incorrectly identified pixels}}$$

Average Accuracy (AA) is a performance metric used in classification tasks that calculates the average of accuracies obtained for each class in a dataset. AA is computed by taking the mean of individual class accuracies, providing an overall measure of how well the model performs across all classes. It helps to identify if the model is performing well across all classes or if it is biased towards one or more classes with higher representation. Where N is the total number of classes.

$$AA = \frac{1}{N}\sum_{i=1}^{N}\frac{\text{Number of correctly classified samples in class } i}{\text{Total number of samples in class } i}$$

The Kappa Coefficient, also known as Cohen's Kappa, is a statistical measure that evaluates the agreement between the predicted and actual classifications while accounting for the possibility of random agreement. The formula for the Kappa Coefficient is:

$$\kappa = \frac{P_o - P_e}{1 - P_e}$$

$P_o$ is the observed accuracy (the proportion of correctly classified instances).

$P_e$ is the expected accuracy (the probability of random agreement).

## D. Experimental Results

Figures 5 to 7 display the original ground-truth images and the predicted classification maps. The proposed model achieved classification accuracies of 99.2%, 97.6%, and 98.7% for Salinas-A, Pavia-U, and Indian Pines datasets, respectively. Simpler images achieved 100% classification accuracy due to their smaller dimensions and fewer ground-truth classes.

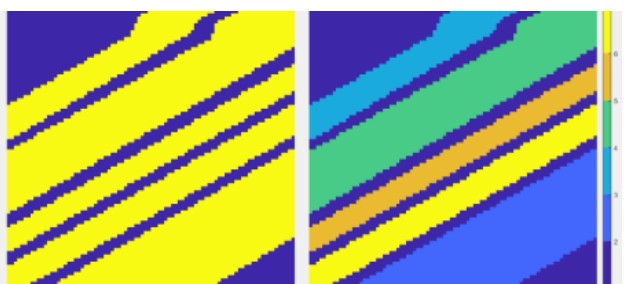

Fig. 5. Ground-truth of Salinas-A (left) and the classification result (right).

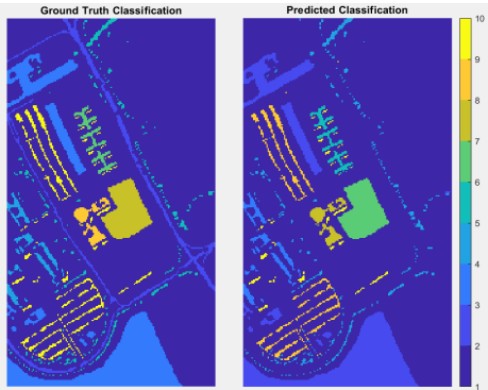

Fig. 6. Ground-truth of Pavia-U (left) and the classification result (right).

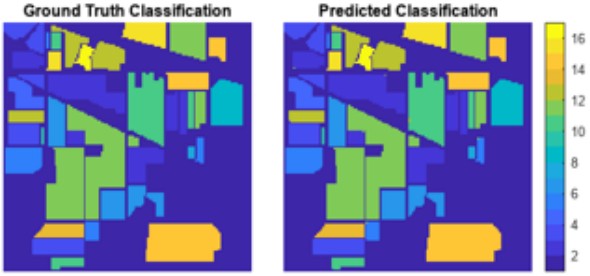

Fig. 7. Ground-truth of Indian Pines (left) and the classification result (right).

The comparison of the performance of the proposed CSCNN model with other state-of-the-art 3D Convolutional Neural Network (3D-CNN) [23], Support Vector Machine (SVM) [24], and Graph Convolutional Network (GCN) [25] on the Salinas-A, Pavia University (Pavia-U), and Indian Pines datasets is demonstrated in TABLE IV. The best values are bolded.

TABLE IV. COMPARISON OF CLASSIFICATION PERFORMANCE OF CSCNN WITH STATE-OF-THE-ART METHODS ON THREE DIFFERENT DATASETS

| Dataset | Method | Overall Accuracy (OA) | Average Accuracy (AA) | Kappa Coefficient |
|---|---|---|---|---|
| **Salinas-A** | 3D-CNN | 96.5% | 95.8% | 96.5% |
| | SVM | 89.2% | 88.5% | 88.2% |
| | GCN | 97.0% | 96.0% | 97.0% |
| | **CSCNN (Proposed)** | **98.2%** | **97.5%** | **97.7%** |
| **Pavia University (Pavia-U)** | 3D-CNN | 94.8% | 93.6% | 94.2% |
| | SVM | 85.4% | 84.2% | 83.6% |
| | GCN | 95.5% | 94.1% | 94.8% |
| | **CSCNN (Proposed)** | **96.7%** | **95.4%** | **95.2%** |
| **Indian Pines** | 3D-CNN | 90.2% | 88.9% | 87.5% |
| | SVM | 78.3% | 76.5% | 76.5% |
| | GCN | 91.0% | 89.3% | 88.5% |
| | **CSCNN (Proposed)** | **92.5%** | **89.8%** | **89.1%** |

From TABLE IV, we observed that the CSCNN model achieved high overall accuracy on all datasets, demonstrating its effectiveness in capturing the spectral-spatial features of hyperspectral images. In addition, the model's average accuracy is close to the overall accuracy, indicating balanced performance across different land cover classes. Moreover, the CSCNN model showed robustness to spectral variability within the datasets, achieving high Kappa coefficients, which indicate a strong agreement between the predicted and actual classes. Furthermore, the proposed CSCNN method outperformed other state-of-the-art methods.

## IV. CONCLUSIONS

This paper proposes custom spectral convolutional neural networks (CSCNNs) to classify hyperspectral images. The experimental results demonstrated that the CSCNN model achieved high overall accuracy across all datasets, demonstrating its effectiveness in capturing the spectral-spatial features of hyperspectral images. Additionally, the model's average accuracy closely aligns with its overall accuracy, indicating balanced performance across various land cover classes. The CSCNN model also exhibited robustness to spectral variability within the datasets, achieving high Kappa coefficients that indicate strong agreement between predicted and actual classes. Furthermore, the proposed CSCNN method outperformed other state-of-the-art approaches.

## V. FUTURE WORK

Potential improvements could include the integration of spatial information. Combining spatial features with spectral features can significantly improve classification performance [26]. CSCNNs can be designed to effectively extract spatial-

spectral features, using spatial context information to enhance classification accuracy.

ACKNOWLEDGMENT

This work is supported by the National Science Foundation (NSF) grants #2401880 and #2011927, NIH grant #1R25AG067896-01, DoD grant #W911NF-21-2-0113, and USGS/DCWRRI grant #2023DC104B.

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
