# OpenReview forum: "Hyperspectral Image Classification Using Custom Spectral Convolutional Neural Networks (CSCNNs)"
_IEEE.org/ICIST/2024/Conference — IEEE ICIST 2024 Conference Submission_

### Official Review · Reviewer_s6jc · 2024-09-02
**Accept**

**Rating:** 7
**Confidence:** 2

**Review:**

This paper proposes a hyperspectral image classification method based on CSCNNs. The method leverages the high-dimensional spectral data inherent in hyperspectral images, employing specially designed convolutional layers to capture spectral-spatial features. Extensive experiments on relevant datasets validate the performance of CSCNNs, achieving higher classification accuracy and stronger performance metrics. The paper is innovative and well-structured, but the following comments could further enhance its quality:
1.The paper has some noticeable formatting issues, such as inconsistencies in the identification of the corresponding author and improper alignment of the first line of the text.
2.The literature review is relatively sparse.
3.Table 1 is too large, and Figure 1 has a low resolution.
4.There is excessive white space on the fifth page.

---

### Official Review · Reviewer_NEn6 · 2024-09-02
**Hyperspectral Image Classification Using Custom Spectral Convolutional Neural Networks (CSCNNs)**

**Rating:** 7
**Confidence:** 2

**Review:**

This paper proposes a CSCNNs-based approach to classify hyperspectral images. The experimental results demonstrated that the CSCNN model achieved high overall accuracy across all datasets, demonstrating its effectiveness in capturing the spectral spatial features of hyperspectral images. The results in this paper is new and interesting. Some minor suggestions are proposed below. 1) Please highlight the main contributions for reading. 2) The motivation of this paper should be clarified.

---

### Official Review · Reviewer_F1dX · 2024-09-02
**Accpet**

**Rating:** 7
**Confidence:** 5

**Review:**

This paper investigates Custom spectral convolutional neural networks (CSCNNs) subject to image classification. After my review of this article, this article can be published in its current form. But there are still some suggestions:
1. There seems to be a formatting problem in the first paragraph of the introduction.
2. This paper presents several advantages over traditional pixel-based approaches. The authors can represent them more concisely.

---

### Decision · Program_Chairs · 2024-09-08

Accept (Oral)